# Neuroimaging of Sex/Gender Differences in Obesity: A Review of Structure, Function, and Neurotransmission

**DOI:** 10.3390/nu12071942

**Published:** 2020-06-30

**Authors:** Danielle S. Kroll, Dana E. Feldman, Catherine L. Biesecker, Katherine L. McPherson, Peter Manza, Paule Valery Joseph, Nora D. Volkow, Gene-Jack Wang

**Affiliations:** 1Laboratory of Neuroimaging, National Institute of Alcohol Abuse and Alcoholism, Bethesda, MD 20814, USA; danielle.kroll@nih.gov (D.S.K.); dana.feldman@nih.gov (D.E.F.); catherine.biesecker@nih.gov (C.L.B.); katherine.mcpherson@nih.gov (K.L.M.); peter.manza@nih.gov (P.M.); paule.joseph@nih.gov (P.V.J.); nora.volkow@nih.gov (N.D.V.); 2Sensory Science and Metabolism Unit, National Institute of Nursing Research, Bethesda, MD 20814, USA; 3Office of the Director, National Institute of Drug Abuse, Bethesda, MD 20814, USA

**Keywords:** obesity, sex, gender, taste, neuroimaging, MRI, PET, opioid, dopamine, serotonin

## Abstract

While the global prevalence of obesity has risen among both men and women over the past 40 years, obesity has consistently been more prevalent among women relative to men. Neuroimaging studies have highlighted several potential mechanisms underlying an individual’s propensity to become obese, including sex/gender differences. Obesity has been associated with structural, functional, and chemical alterations throughout the brain. Whereas changes in somatosensory regions appear to be associated with obesity in men, reward regions appear to have greater involvement in obesity among women than men. Sex/gender differences have also been observed in the neural response to taste among people with obesity. A more thorough understanding of these neural and behavioral differences will allow for more tailored interventions, including diet suggestions, for the prevention and treatment of obesity.

## 1. Introduction

The global prevalence of overweight (body mass index (BMI) > 25 kg/m^2^) and obesity (BMI > 30 kg/m^2^) has risen from 24.6% in 1980 to over one-third of the world’s population in 2015 [1]. Although this pattern has been seen in both sexes/genders, obesity is more common in women relative to men, independent of age, geographic region, or socioeconomic status [1]. The deleterious effects of obesity on many of the body’s organ systems in both sexes/genders have been well documented. Individuals with obesity are more likely to develop type 2 diabetes mellitus [2], cardiovascular disease [3], certain types of cancer [4], musculoskeletal disorders [5], and psychiatric illness [6,7]. However, obesity can present differently in women than in men. Premenopausal women tend to have a higher subcutaneous to visceral fat ratio due to their high levels of estrogen. This pattern of fat distribution has been shown to protect against some metabolic complications [8]. Nevertheless, given the wide-spread adverse effects of obesity, it is important to understand the disparate prevalence of obesity among men and women.

Neuroimaging can elucidate aspects of brain structure, function, and chemistry that are associated with sex/gender differences in compulsive eating behaviors in obesity. Several groups have linked the neural mechanisms underlying obesity to those for substance use disorders (SUDs) [9,10,11]. In obesity, palatable food consumption activates the mesolimbic dopaminergic pathway, begetting its rewarding and conditioning effects, as is the case for drug consumption in SUD. Repeated activation of this pathway in response to palatable food intake causes neuroadaptations in downstream inhibitory cortico-striatal circuits, leading to compulsive food intake [12]. While both sexes/genders are believed to share this common neuroadaptation, male and female individuals exhibit unique neural signatures in response to food cues (recently reviewed in Chao et al., 2017 [13]) and taste cues [14,15]. Women demonstrate greater neural responses in striato-limbic and frontal-cortical regions in response to food cues than men, perhaps underlying the elevated prevalence of obesity among women [13].

Sex/gender differences also present behaviorally in populations with obesity. Several behavioral studies have indicated that men and women have different food preferences. Women are more likely to prefer sweet tastes, high-calorie foods, and snack-based comfort foods, such as candy, whereas men tend to prefer savory tastes, low-calorie foods, and meal-based comfort foods, such as pizza [16,17,18]. Further, men and women show different patterns when engaging in dieting efforts, with women being more likely to report a history of dieting than men [19]. Sex/gender differences in dieting are also linked to societal norms that impact self-perception [20,21]. Women and men perceive their desired weight differently, and women tend to display lower self-esteem and higher levels of dieting than men [20,21]. Diet interventions among women have further been associated with increased activation of cortical areas involved with inhibitory control and self-regulation following food consumption, perhaps normalizing brain function in obesity [22]. However, to our knowledge, no studies have been done on the neural response of diet interventions among men, likely due to the lower frequency of dieting among men [23]. As such, a better understanding of the neural underpinnings of sex/gender behavioral differences in obesity can be used to devise more effective lifestyle interventions.

We recognize that there are many semantic ambiguities in the field of obesity and sex/gender difference research. Here, we digress briefly to clarify some important distinctions between commonly used terms and definitions. For the purposes of this review, as is common in the existing literature, we focus on obesity, which is defined by BMI. However, BMI, and other common metrics used to describe obesity, including total body fat, are not strong predictors of obesity’s metabolic comorbidities. The ratio of subcutaneous to visceral fat, while outside of the scope of this review, is a more appropriate indicator of these complications (reviewed in [8]). We further acknowledge that obesity is highly comorbid but not synonymous with binge eating disorder (BED) and food addiction, and that those conditions also exist in lean individuals [9]. In this review, we also reference sex/gender, as we recognize that a difference exists between the concepts of biological “sex” and socio-cultural “gender” [24]. For a more thorough discussion on the use of this term, see Chao et al. (2017), who highlight the practical complexities of disentangling these two distinct, yet related, concepts [13].

In this paper, we review multimodal neuroimaging findings as they relate to sex/gender differences in obesity. We describe the differences in brain structure, resting-state function, and neurotransmission that exist between male and female individuals with obesity. We also summarize the sex/gender differences in the neural response to taste cues. These findings are tabulated in Tables 1–4 and structural and functional changes are further demonstrated visually in Figures 1 and 2. After integrating these results, we discuss the evidence for the development of sex/gender-specific diet interventions for individuals with obesity.

## 2. Structure

Several studies have examined gray matter volume (GMV) differences in men and women with obesity, as outlined and illustrated in Table 1 and Figure 1, respectively. One study found a positive association between GMV and BMI in the right orbitofrontal cortex (OFC) and nucleus accumbens (NAcc) in both men and women; GMV also positively correlated with central leptin levels, another positive marker of obesity [25]. However, in women, but not men, BMI and central leptin also positively correlated with GMV in the left putamen, and central leptin levels negatively correlated with GMV in the right dorsolateral prefrontal cortex [25]. These findings suggest a link between obesity and greater GMV in reward-associated regions that appears to be stronger among women. This structural difference perhaps could underlie the greater preference for immediate rewards despite long-term negative consequences among obese versus lean women compared to that between obese and lean men [25]. In another study among a large cohort of lean, overweight, and obese adults, both sexes/genders demonstrated a negative correlation between total body fat and GMV in the globus pallidus. In men, but not women, total body fat was also negatively associated with GMV in subcortical regions, including the thalamus, caudate nucleus, putamen, hippocampus, and NAcc [26] These associations between obesity metrics and GMV in brain reward regions, which may be specific to each sex/gender, might contribute to some of the behavioral manifestations associated with sex/gender differences in obesity. Due to the cross-sectional nature of these studies, the temporal relationship between GMV and obesity remains unknown. 

Obesity also appears to alter white matter (WM) structure and organization in a sex/gender-dependent manner. Many measures derived from diffusion tensor imaging provide insight into this relationship, including axial diffusivity, a measure of axonal integrity; radial diffusivity, a negative measure of myelination; and fractional anisotropy, a measure of WM microstructural integrity [27]. One study of individuals with obesity found that BMI negatively correlated with axial diffusivity in the corpus callosum for both sex/genders [28]. In females only, BMI and serum leptin levels also positively correlated with radial diffusivity and negatively correlated with fractional anisotropy in the corpus callosum [28]. However, Dekkers et al. (2019) found that in women, but not men, total body fat negatively associated with global mean diffusivity, a measure negatively associated with WM integrity [26]. Together, these findings suggest a stronger relationship between obesity and WM integrity in women than in men, though the direction remains unclear. One limitation of these findings is that, to our knowledge, only Dekkers et al. (2019) controlled for total brain volume or intracranial volume; recent studies have shown that when these measures are accounted for, many regional sex/gender differences disappear [29,30].

Age may also interact with sex/gender and obesity in the brain structure. Ronan et al. (2016) demonstrated that participants who were obese or overweight displayed a lower cerebral WM volume, a finding typically observed in normal aging [31]. This effect was independent of sex/gender and was age dependent, with the greatest volume loss, adding an estimated 10 years of ‘brain age’, occurring at around age 40 [31]. A longitudinal study among women with obesity corroborated this finding by demonstrating that women were more likely to experience atrophy of the temporal lobe as both BMI and age increased [32]. However, not all studies on brain volume in people with obesity have been consistent. In one study, obesity was positively associated with frontal and temporal WM as well as hippocampal volume among women aged 70–89 [33]. Other studies similarly found that among women, but not men, obesity appears to be protective against GMV loss through aging, slowing down both hippocampal volume decline and ventricular enlargement [34]. Finally, sex/gender differences in obesity seem to only arise when participants reach a certain age. Children with obesity with a mean age of 9.0 years (range: 9.0 ± 0.9 years) displayed no significant sex/gender differences in brain structure, although obese children exhibited lower GMV than lean children in the temporal lobe, dorsolateral and medial prefrontal cortices, and the right anterior cingulate cortex [35]. Preclinical studies seem to yield opposite results; at a young age, male mice were more susceptible to the negative effects of a high-fat diet, such as decreased WM integrity and cerebral blood flow, than female mice [36]. Age, therefore, seems to interact with sex/gender and obesity, but this relationship is not fully understood.

Limited evidence suggests that low-calorie diets can contribute to brain structure recovery in a sex/gender-dependent manner. A longitudinal study that did not report sex/gender differences indicated that at baseline, participants with obesity have greater WM volumes in the temporal lobes, brainstem, and cerebellum [37]. After a 6-week low-calorie diet, these participants demonstrated that this expansion of WM volume partially recovered [37]. Some preclinical studies, though, have shown that low-calorie diets can affect these brain changes in a sex-specific manner. For instance, in control mice, levels of proliferating cells and neuroblasts are significantly higher in females than in males. However, when both sexes eat a high-fat diet, this difference disappears, suggesting that a high-fat diet contributes to a reduction in adult hippocampal neurogenesis in females only [38]. These preclinical results have yet to be replicated in humans. Nonetheless, low-calorie diets appear to contribute to structural brain changes in obesity. 

## 3. Function

### 3.1. Resting State

It is well established that sex/gender differences exist in human brain functional connectivity [39,40]. Elevated BMI, the most common marker of obesity (BMI > 30 kg/m^2^), has also been associated with brain connectivity [41,42]. However, the interaction between sex/gender and obesity remains unclear. Resting state connectivity in functional magnetic resonance imaging (fMRI) is an important but understudied area as it pertains to sex/gender differences in obesity. Few studies by one group of researchers [43,44,45] provide a preliminary understanding of the effects of sex/gender and BMI on brain connectivity, though these results, to our knowledge, have yet to be replicated by others. These results are illustrated in Figure 2 and outlined in Table 2.

There are many methodological approaches to examine resting state fMRI signals, but existing studies have focused on network centrality and slow-wave connectivity. Centrality generally assesses the connectedness of a given brain region to many others [46]. Gupta et al. (2017) found that women with obesity have higher centrality measures in several regions of the reward network, including the left amygdala, right NAcc, and bilateral hippocampus, than men with obesity, while men have higher centrality measures in the bilateral putamen [43]. Measures of food addiction also seem to contribute to the centrality of reward regions among women; scores on the Yale Food Addiction Scale, a standardized and widely used metric to quantify food addiction, were positively associated with the centrality of the ventral tegmental area (VTA) among women whereas they were negatively associated among men [45,47]. While men with obesity have lower centrality in reward regions than their female counterparts, Gupta et al. (2017) demonstrated that men with obesity have greater centrality in the right putamen, hippocampus, and medial orbitofrontal gyrus relative to lean men, and women with obesity have greater centrality in the left amygdala than lean women [43]. Thus, alterations in the centrality of the reward network tend to occur in a sex/gender-dependent manner, with female individuals showing greater centrality than their male counterparts. However, the interaction between sex/gender and obesity on brain network centrality has yet to be formally tested, so it remains unclear.

Obesity also shows a correlation with slow-wave connectivity throughout the reward network in a sex/gender-dependent manner. Another study by Gupta et al. (2018) examined two classes of slow-wave neural signals, labelled slow-4 and slow-5, in obese and lean men and women [44]. The slow-4 signal (medium frequency) is thought to arise from the basal ganglia, while the slow-5 signal (low frequency) is thought to arise from the cortex [48]. The slow-4 signal in the right globus pallidus and bilateral putamen was associated with BMI in females but not in males [44]. Further, among females, higher BMI was associated with lower slow-5 connectivity between the left globus pallidus and substantia nigra with the bilateral posterior mid cingulate cortex and frontal cortical regions [44]. This finding is reminiscent of the aberrant cortico-striatal signaling characteristic of obesity and substance use disorders [12,44]. Conversely, among males, greater BMI was associated with higher left globus pallidus and substantia nigra slow-5 connectivity with the medial frontal cortex [44]. Centrality measures have corroborated these findings; in women, higher food addiction scores were associated with lower centrality in frontal areas and higher centrality in VTA and this pattern is reversed in men [45]. This further suggests that neuroadaptations in reward regions appear to play a larger role in compulsive eating in women than for men.

### 3.2. Taste Response

Sex/gender differences in neuroimaging taste perception in people with obesity are an underexplored research area. This is partially due to the difficulty in firmly identifying regions in the human brain associated with taste (recently reviewed in Kure Liu et al., 2019 [49]). It has been established that the anterior insula/frontal operculum is the primary taste cortex [50,51,52,53,54]. However, Avery et al. (2020) reported that taste quality is more accurately analyzed using a combinatorial spatial code, in which taste perception is distributed throughout the sensory network [55]. In this model, taste quality refers to the unique pattern of activation throughout the primary taste cortex and regions involved in processing hedonic and aversive tastes for each individual in a population [55]. This contrasts with a topographic perspective in which activated regions are attributed to each separate taste component [55]. The type of taste (i.e., salty, bitter, sweet, sour, or umami) has also been found to influence brain activation differences between sexes/genders [15]. The diversity of responses to taste prevent researchers from finding a specific and reliable way to pinpoint regions activated by taste using neuroimaging.

Despite these existing limitations, related studies can shed light on neural taste perception in men and women with obesity (Table 3). Specifically, sex/gender differences in anticipation of taste may improve our understanding of underlying neural differences leading to obesity. For example, Cornier et al. (2015) examined taste anticipation in men and women who were identified as obese prone or obese resistant based on a history of diet and weight gain but not current BMI [56]. After undergoing a cue reactivity fMRI task associating sucrose and artificial saliva with visual cues, males in both populations displayed greater neuronal response to the sucrose-associated visual cue in the right caudate nucleus relative to women [56]. The study supports that there are brain activation sex/gender differences in anticipation of receiving sugar but not the receipt of the sugar itself, emphasizing the importance of sex/gender differences in conditioning in obesity. Geliebter et al. (2013) presented a related analysis of sex/gender differences in obese individuals [57]. When presented with high- instead of low-energy dense auditory food cues, male participants with obesity portrayed brain activation in supplementary motor areas (precentral gyrus) in a sated state. Female participants, though, showed activation in cognitive-related regions (parahippocampal gyrus) in a fasted state [57]. The same obese population further demonstrated that otherwise healthy men and women displayed different patterns of functional connectivity in the amygdala and ventral striatum when responding to food cues in both a sated and hungry state [58]. When the subjects were in a sated state, men tended to show greater connectivity in the amygdala than women, while women displayed greater connectivity in the angular gyrus and precentral gyrus than men [58]. However, in the fasted state, the motor/visual processing centers and emotion/reward-related regions (supplementary motor area, precentral gyrus, precuneus, cuneus) in men were more highly connected, while women had greater connectivity in areas involving response inhibition and cognitive control (i.e., inferior frontal gyrus). Atalayer et al. (2014) suggest their results support the hypothesis that men may process hunger in relation to emotional cues, while women relate it to cognitive processing. The combination of findings from these studies illustrate how satiety and food anticipation may impact brain activation differently in men and women [56,57,58].

Preliminary evidence suggests that bariatric surgery can impact taste reward anticipation in both sex/genders. In one study, 13 patients with obesity who had undergone a gastric bypass experienced an expected weight loss, in addition to changes in the neural response to the expectation of tastants from before surgery to one-month post-operative [59]. Anticipation of sweet and salty stimuli evoked responses in reward regions, including the NAcc, caudate, VTA, OFC, and prefrontal cortex, as measured by fMRI [59]. While this neural response decreased from baseline to 1-month post-operative in anticipation of sucrose, it increased in anticipation of sodium chloride. 

However, lean control participants, who did not have a gastric bypass and were scanned 1 month apart also had a similar decrease in reward response to sucrose but no change in response to sodium chloride, and so it is unclear if this change in sucrose anticipation is due to habituation or the gastric bypass [59]. Some sex/gender differences have also been identified regarding changes in taste response following bariatric surgery. Another group found men (*n* = 35) exhibited a greater decrease in taste and smell ability than women (*n* = 120) as assessed using a subjective taste questionnaire five years post-sleeve gastrectomy operation, especially among those with type-2 diabetes [60]. The authors did not report sex/gender differences in starting BMI, nor what type of taste changes the subjects underwent. Further, there were no sex/gender differences following roux-en-Y gastric bypass in taste or taste aversion [60]. Since these are pilot results, their replication as well as clarification of what types of taste changes occur following bariatric surgery and how they differ between sexes/genders are still needed.

Research in a lean population analyzing sex/gender differences in taste also contributes valuable insight on how weight and taste interact between sexes/genders. While maintaining insignificant differences in BMI (23.15 kg/m^2^ average for men and 22.76 kg/m^2^ average for women), one cohort displayed notable sex/gender differences in the neural response to the transition from hunger to satiety to four different taste types: Sour, bitter, sweet, and salty [15]. fMRI results indicated greater brain activation decreases in men compared to women in the middle frontal gyrus, insula, and cerebellum when changing from a hunger to satiety state for all four tastes [15]. The middle frontal gyrus has been identified as an area critical to dual-task performance and decision-making [61,62]. Since women demonstrated consistently high activation through both hunger and satiety while men experienced a decrease in activation following satiety, Haase et al. (2011) speculated that women exhibit greater top-down functioning regarding taste salience [15]. In other words, female brains may process taste input more cognitively than males’ brains since they were activated even when sated. In comparison, male brains displayed less activation after feeling full, suggesting their taste processing is built up from a small piece of sensory information, which dissipates after it has been resolved. Compared to women, men also displayed greater activation changes in reaction to sucrose, citric acid, and caffeine in the inferior frontal gyrus; sucrose and NaCl within the parahippocampal gyrus, entorhinal cortex, perirhinal cortex, and amygdala; and sucrose within the dorsal striatum (caudate, putamen) and posterior cingulate [15]. Because these areas are involved in reward and memory processing, these findings imply that women encode and learn about their food differently than men [63,64,65]. However, how these neural differences play out behaviorally in obese and overweight populations remains unclear.

Another study examined first-year college students and found that weight gain over the first eight months of school was associated with concurrent taste changes: For every 1% body weight increase, male students displayed a decrease in their tasting ability for sweetness (by ≤ 11.0%) and saltiness (by 7.5%) [66]. This finding aligns with similar studies that observed a decrease in sweet and salty perception accompanying weight gain in male more than in female students [67,68]. In comparison, female students did not display this taste decrease, and instead displayed a 6.5% increase in sourness perception ability for every 1% gain in body weight [66]. However, this correlational study cannot conclude a causal relationship between weight gain and changes in taste perception. While the authors suggested that taste differences between sexes/genders are due to hormonal effects, the neural findings from Haase et al. (2011) suggest that functional differences in the limbic system also contribute to differences in taste perception [15,66,69]. 

## 4. Neurotransmission

### 4.1. Serotonin Signaling

Serotonin (5-HT) is involved with regulation of appetite and has shown aberrant signaling in animal models of obesity [70,71]. 5-HT signals through several subtypes of 5-HT receptors, including the serotonin 1A (5-HT1A) and serotonin 2A (5-HT2A) receptors. The 5-HT1A receptor has widespread inhibitory actions, with autoreceptor negative feedback function in some cells and postsynaptic distribution in others (reviewed by Carhart-Harris and Nutt, 2017) [72]. The 5-HT2A receptor has postsynaptic distribution throughout the brain, with generally excitatory action [72]. To date, no clinical positron emission tomography (PET) or single-photon emission computerized tomography (SPECT) studies have assessed 5-HT1A receptors in obesity; however, preclinical studies provide insight into a relationship between 5-HT1A and food intake [73,74]. 5-HT signaling has also been associated with sex hormones in rats [75], but clinical imaging studies have shown conflicting results, with Jovanovic et al. (2008) and Parsey (2002) finding higher radiotracer binding to 5-HT1A in women, and Moses-Kolko et al. (2011) finding higher binding in men [76,77,78]. Clinical studies of sex/gender differences in obese individuals are needed.

Sex/gender differences in the 5-HT2A receptor have also been investigated (Table 4). Although positive correlations between binding of the 5HT2A receptor [^18^F]altanserin and estradiol levels have been shown in men [79] and between [^18^F]deuteroaltanserin binding and estradiol levels in women [80,81], sex/gender differences have not been observed [77,82,83,84,85,86]. To our knowledge, no PET or SPECT studies have compared 5-HT2A receptor binding between obese and normal-weight individuals, though one study reported a positive correlation between BMI and 5-HT2A receptor binding [82]. Future studies considering obesity, sex/gender, and 5-HT2A receptor binding are needed.

Sex/gender differences in serotonin transporter (5-HTT) availability have shown inconsistent results among normal-weight participants [76,87,88]. Additionally, not much is known about the relationship between obesity and 5-HTT. While 5-HTT availability was found to be negatively correlated with BMI [89] in one [^11^C]3-amino-4-(2-dimethylaminomethyl-phenylsulfanyl)benzonitrile (DASB) PET study, a [^123^I]2beta-Carbomethoxy-3beta-(4-iodophenyl)nortropane (nor-β-CIT) SPECT study did not demonstrate this [90]. This could be explained by differences between tracer and imaging modalities; both tracers are selective for 5-HTT, but [^123^I]nor-β-CIT is also selective for the dopamine transporter in the striatum [91], whereas [^11^C]DASB is selective for 5-HTT in the striatum [92]. Further, PET and SPECT differ in spatial resolution [93]. Female monozygotic twins with higher BMIs had higher [^123^I]nor-β-CIT binding in the hypothalamus and thalamus than their leaner co-twins. This was not observed in male twinsets, suggesting obesity may have a sex/gender-dependent association with 5-HTT availability [90]. Further, [^123^I]nor-β-CIT SPECT studies of BED may elucidate how obesity and 5-HTT availability are related: Compared to control women with obesity, women with obesity and BED had lower 5-HTT binding [94]. In an intervention study, women with obesity and BED showed improved midbrain 5-HTT binding during fluoxetine, a selective serotonin reuptake inhibitor used for weight loss in individuals with obesity [95] and therapy-facilitated remission [96]. Control women with obesity who did not receive fluoxetine did not show a change in 5-HTT binding [96]. These studies suggest that BED may be driving 5-HTT binding differences in other studies assessing obesity, assuming that fluoxetine administration would not benefit women without BED.

### 4.2. Dopamine Signaling

Dopamine (DA) signaling energizes the motivation towards food and preclinical studies have associated DA signaling dysregulation with obesity [97,98,99]. Clinical brain imaging studies focusing on sex/gender differences in DA D2 and D3 receptor availability have yielded mixed results [100,101,102,103]. In obese subjects, brain imaging studies reported that binding of [^11^C]raclopride, a tracer selective to D2 and D3 receptors, in the striatum was lower in participants with obesity compared to controls [104,105,106], suggesting downregulation of D2/D3 receptor availability in obesity. Similar findings were observed in overweight and obese participants (BMI > 27 kg/m^2^) compared to controls [107]. There are sex/gender differences in both BMI and D2/D3 receptor availability, suggesting there may be a common underlying phenotype driving these associations (Table 4). Women have a higher incidence of obesity, and tend to have lower D2/D3 availability, than men on average [100], and lower D2/D3 availability has been independently associated with high BMI [104,105,106,107]. Further, other studies, however, did not demonstrate differences between these groups [108], including one study utilizing the D2 receptor radiotracer N-[^11^C]-methyl-benperidol [109]. Several of these studies either assessed female-only or mixed-sex/gender samples and lacked the power to investigate sex/gender differences. In one study that did take sex/gender into account, significant sex/gender differences were not observed, although there were only 10 individuals with obesity in that sample [106]. Studies with larger sample sizes are needed to elucidate whether DA signaling in obesity is sex/gender dependent. 

In terms of weight-loss interventions, not much is known about their impact on D2/D3 receptor availability. To our knowledge, only three studies to date have assessed gastric bypass surgery. Each study used all-female cohorts and had different findings, with one study showing increased [^11^C]raclopride binding, another showing decreased binding, and the third showing no significant difference from baseline after the surgery-induced weight loss [110,111,112]. Due to the inconsistency in results, more research is needed on surgery-induced weight loss, as well as other interventions, to determine whether these treatments impact the DA system. Men should also be included in these investigations to determine whether sex/gender plays a role in weight loss-related changes in DA signaling.

Findings of sex/gender differences in DA release are also mixed [101,103]. In a [^123^I]iodobenzamide SPECT study, female controls showed significant DA release in response to amphetamine, while women with severe obesity did not show a significant change from baseline [104]. Further, BMI and DA release in response to a caloric glucose stimulus (compared to calorie-free sucralose) were negatively correlated [113]. These findings supported a disruption of DA signaling in individuals with high BMI. In another study, however, DA release after glucose injection (compared to saline) did not differ between participants with BMIs above 27 kg/m^2^ and lean controls [107]. It is possible that the conflicting results were due to the difference in the route of administration or that BMI alone may not predict differences in DA release. Another study by Wang et al. (2011) comparing women with obesity to women with obesity and binge eating disorder (BED) showed that those with BED had enhanced DA release to a food stimulus [114]. Thus, binge eating might drive the previous findings discussed. However, van de Giessen et al. (2014) ruled out BED from their group with obesity and still observed differences between women with obesity and normal-weight controls, so more studies are necessary to determine the relationship between obesity and BED with DA release [104]. These opposing findings may be linked to the difference in displacement patterns between [^123^I] iodobenzamide and [^11^C]raclopride. However, the two radiotracers have shown similar patterns of displacement in the past (as reviewed by [115]); [^123^I]iodobenzamide the mono-iodine analog of [^11^C]raclopride [116]. Further, no studies have yet assessed sex/gender differences in obesity regarding the impact of weight-loss interventions on DA release, which highlights the need for broadened investigations into this area.

Finally, sex/gender differences in the availability and distribution of DA transporter have not been observed [117,118,119], nor have these studies shown differences between individuals with obesity and controls [118,119] in SPECT studies. However, these studies still need to be replicated, especially because each study cited used a different tracer (Nam et al. (2018) used [^123^I]FP-CIT, Thomsen et al. (2013) used [^123^I]PE2I, and Best et al. (2005) used [^123^I]βCIT) [117,118,119]. The pharmacokinetics and pharmacodynamics differ between these radiotracers; [^123^I]PE2I, for example, is faster and has higher affinity for DAT than [^123^I]FP-CIT and [^123^I]βCIT [120].

### 4.3. Opioid Signaling

Opioid signaling has been implicated in obesity by interacting with other neurotransmitters to regulate feeding and satiety [124]. Sex/gender differences in molecular imaging studies of mu, kappa, and delta opioid signaling are unclear, as very few studies have compared men to women [125,126,127]. Opioid signaling in obesity may be altered: In two all-female [^11^C] carfentanil PET studies, obesity was associated with decreased mu-opioid receptor (MOR) availability throughout the brain (see Table 4) [108,122,123]. Further, Tuominen et al. (2015) found a correlation between [^11^C] carfentanil and [^11^C]raclopride binding in the ventral striatum and dorsal caudate nucleus in lean control women but not in the ventral striatum of women with obesity [123]. This suggests alterations in the link between MOR and D2/D3 receptors in the ventral striatum of women with obesity. In a study with all men, MOR availability was lower in the temporal pole, amygdala, prefrontal cortex, and thalamus in obese participants compared to controls [121]. After a restricted calorie intervention, MOR availability among this male cohort partially recovered in the left temporal pole, ventral striatum, thalamus, and medial frontal cortex [121]. Another PET study with a cohort of men and women, however, did not observe an association between BMI and binding of the non-selective opioid receptor radioligand [^18^F] FDPN, though sex/gender differences were not assessed [125]. Given the differences in MOR, this negative finding may indicate a contribution of kappa and delta opioid receptors in obesity. Interestingly, mice without delta opioid receptors were shown to have resistance to diet-induced obesity [128]. More consistent studies of opioid signaling in people with obesity could help clarify these discrepancies.

## 5. Conclusions

This review aimed to describe the neural underpinnings of sex/gender differences in obesity. In general, obesity is associated with an abnormal structure, function, and chemistry in the brain’s reward system [12]. This is characterized by a smaller volume in the NAcc, OFC, and globus pallidus and downregulation of D2/D3 receptors in the striatum [25,26,104,105,106,107]. Women appear to be more susceptible to neural adaptations associated with obesity than men [25,26,28,34,43,44,45,90,123]. Women with obesity also tend to have a greater volume and centrality measures in subcortical reward regions and lower volume and centrality measures in frontal cortical regions [25,26,28,34,43,44,45]. Men with obesity, however, seem to have more effects in cortical somatosensory regions, the putamen, and thalamus [26,43]. These findings are consistent with activation patterns in response to food cues among men and women with obesity (reviewed by Chao et al., 2017) and suggest distinct neural mechanisms in obesity for each sex/gender [13]. While many of the papers reviewed here interpreted smaller volume as atrophy [31,32,33], it is important to acknowledge that smaller volume measures in participants with obesity can also be indicative of neuroplasticity or genetic predisposition, particularly in younger study subjects; this is an important limitation to consider when interpreting sex differences in brain structure [129].

Moreover, sex/gender differences in the neural correlates of taste perception, diet, and weight-loss treatment can provide insight for the development of new therapies. Results surrounding brain activation in response to tastants have been mixed [15,56,57]. Men with obesity seem to have a greater neural response to high-energy food cues while sated than their female counterparts [57]. However, lean men have a greater decrease in neural response to tastants from the fasted to sated state than lean women [15]. This discrepancy may be explained by an interaction between sex/gender and obesity in the feeling of satiety, though this has not yet been studied. Further, intervention studies among women, including those for bariatric surgery, therapy, and medication, have yielded mixed results in terms of recovery of brain chemistry and taste response [59,60,96,110,111,112]. Similar studies have not been conducted in solely male participants to our knowledge, and so the effect that these interventions have on the male brain remains unclear.

While outside the scope of this review, several endocrine pathways contribute to sex/gender differences in obesity and warrant further research. Preclinical, clinical, and epidemiological studies have demonstrated that estrogen is protective against many metabolic complications associated with obesity (recently reviewed by [130]). As such, the estrogen concentration may in part explain sex/gender differences in the manifestation of obesity and how these differences change with age [130]. However, the role of hormones in modulating dimorphic brain responses is less clear. One study in rats found that insulin and leptin impact feeding behavior in a sex-dependent manner [131]. In male, but not female rats, central administration of insulin led to decreased food consumption. Conversely, in female, but not male, rats, central administration of leptin led to decreased food consumption [131]. These results are consistent with the human studies reviewed here, in which serum leptin was associated with GMV and WM integrity in women but not men [25,28]. Still, animal and human studies of hormonal regulation of the brain are not always consistent. For example, rat models demonstrating the role of sex hormones in 5HT signaling yielded different results than human models [75,76,78,81]. Thus, more work should be done in the field of neuropsychoendocrinology as it pertains to sex/gender differences in obesity to clarify the role of these hormonal pathways.

Though sex/gender differences are a burgeoning area of brain research, many studies lack the power to properly describe them [24]. Many studies reviewed include a sample of only women [32,33,94,96,110,111,112,122,123] or only men [121]. Samples of only women are particularly prominent in studies of eating behavior and weight-loss interventions, which limits our understanding of the generalizability of these treatments across sex/genders [59,60,96,110,111,112]. Moreover, the contribution of BED to these findings has not been very well established. In the studies reviewed, the effect of BED diagnosis on the brain among obese individuals has only been examined among women [94,96,122], despite there being similar prevalence of BED in individuals with obesity of both sex/genders worldwide [132]. Sex/gender differences in the brain have been well described among healthy adults [39,40,100,101,102,103,117,118,119]. Given the different patterns of men and women behaviorally in weight-loss interventions and food intake [16,17,18,19,66], it is imperative to develop a deeper understanding of how these sex/gender differences manifest in the brain in pathological conditions.

## Figures and Tables

**Figure 1 nutrients-12-01942-f001:**
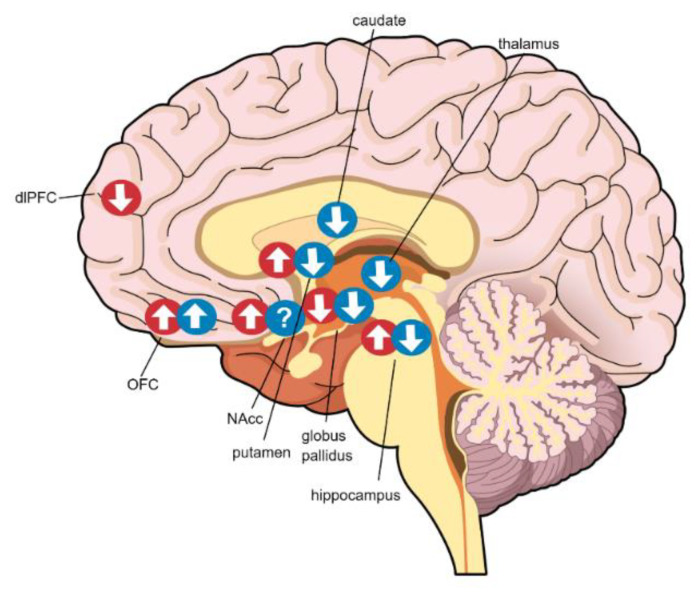
Gray matter volume differences in obesity by sex/gender. Red circles indicate changes among females and blue circles indicate changes among males. Arrows indicate the direction of the correlation between gray matter volume and obesity metrics (i.e., BMI, total body fat, serum leptin). dlPFC: dorsolateral PFC; OFC: orbitofrontal cortex; NAcc: nucleus accumbens.

**Figure 2 nutrients-12-01942-f002:**
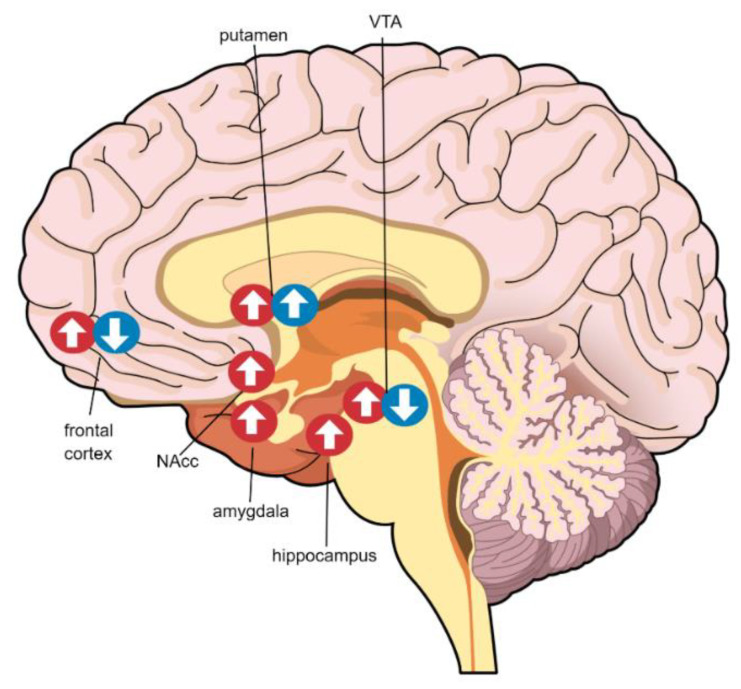
Resting state connectivity differences in obesity by sex/gender. Red circles indicate changes among females and blue circles indicate changes among males. Arrows indicate the direction of the correlation between centrality and obesity metrics (i.e., BMI, total body fat, serum leptin, Yale Food Addiction Scale (YFAS) score). VTA: ventral tegmental area.

**Table 1 nutrients-12-01942-t001:** Brain structure study sample characteristics and findings.

Paper	Age (Years), Mean (SD)	Sample Size	Female, %	Obesity Metrics	Neuro-Imaging Modality	Pertinent Findings
Hortsmann et al. (2011) [25]	Male: 25.46 (4.25) Female: 25.11 (4.43)	122	50	BMI ≥ 30 kg/m^2^ Serum leptin	MRI	Men and women show a positive association between GMV and BMI in the right OFC and NAcc. Women’s BMI and leptin levels positively correlate with GMV in the left putamen; leptin negatively correlates with GMV in the right dorsolateral prefrontal cortex. Women with obesity were more likely to prefer immediate rewards, despite long-term negative consequences, than lean women.
Dekkers et al. (2019) [26]	62.0 (7.3)	12087	52.8	Overweight: BMI ≥ 25 kg/m^2^ Obese: BMI ≥ 30 kg/m^2^ Total Body Fat	MRI & DTI	Men and women show a negative correlation between total body fat and GMV in the globus pallidus. In men, total body fat was also negatively associated with subcortical GMV in the thalamus, caudate nucleus, putamen, hippocampus, and NAcc. In women, total body fat was negatively associated with global mean diffusivity.
Mueller et al. (2011) [28]	26.4 (5.0) Men: 25.5 (5.1) Women: 27.1 (5.0)	49	46.9	BMI ≥ 30 kg/m^2^ Serum leptin	T1w MRI & DTI	In men and women, BMI negatively correlated with axial diffusivity in the corpus callosum. In females only, BMI and serum leptin levels also positively correlated with radial diffusivity and negatively correlated with fractional anisotropy in the corpus callosum.
Ronan et al. (2016) [31]	Lean: 48(16) Overweight: 57(17) Obese: 61 (16)	Lean: 246 Over- weight: 150 Obese: 77	Lean: 49.6 Over- weight: 44 Obese: 63.6	BMI ≥ 30 kg/m^2^	T1w MRI	Greater atrophy of cerebral WM volume in participants who were obese or overweight, independent of sex/gender. This effect was age-dependent, with the greatest atrophy, adding an estimated 10 years of ‘brain age’, occurring at around age 40.
Gustafson et al. (2004) [32]	NR Range: 70–84 years	290	100	BMI ≥ 25 kg/m^2^	CT	Women were more likely to experience atrophy of the temporal lobe as both BMI and age increased.
Driscoll et al. (2016) [33]	NR Range: 65-89 years	1366	100	ND subjects grouped by BMI	T1w MRI	In women aged 70–89, obesity was positively associated with frontal WM, temporal WM, and hippocampal volume.
Armstrong et al. (2019) [34]	71.2 (8.7) Men: 72.2 (8.5) Women: 70.3 (8.7)	617	52.8	BMI ≥ 30 kg/m^2^	T1w MRI	In women, obesity protected against GMV loss as age increased, slowing hippocampal volume decline, and ventricular enlargement.
Xu et al. (2019) [35]	Control: 8.3 (0.9) Prader-Willi: 7.2(1.2) Obese: 9.0(0.9)	Control: 18 Prader-Willi: 12 Obese: 18	66.7	BMI percentile > 95%.	T1w MRI & DTI	No sex/gender differences were found in children with obesity, although subjects with obesity had lower GMV in the temporal lobe, dorsolateral, and medial prefrontal cortices, and the right anterior cingulate cortex.
Haltia et al. (2007) [37]	Lean: 37(21) Obese: 37(12)	Lean: 16 Obese: 30	Lean: 50 Obese: 60	BMI > 27 kg/m^2^	T1w MRI	Both men and women with obesity had greater WM volumes in temporal lobes, brainstem, and cerebellum, but this expansion could recover after a 6-week low-calorie diet.

NR: Not reported. ND: Not defined.

**Table 2 nutrients-12-01942-t002:** Brain resting-state study sample characteristics and findings.

Paper	Age (years), Mean (SD)	Sample Size	Female, %	Obesity Metrics	Neuro-Imaging Modality	Pertinent Findings
Gupta et al. (2017) [43]	30.96 (11.26)	124	50.8	BMI ≥ 25 kg/m^2^	Resting state fMRI	Women with high BMI had higher degrees of centrality in the left amygdala, right NAcc, and bilateral hippocampus than men with high BMI. Men with high BMI have higher centrality measures in the bilateral putamen than women with high BMI. Men with high BMI have greater centrality in the right putamen, hippocampus, and medial orbitofrontal gyrus relative to lean men. Women with high BMI have greater centrality in the left amygdala than lean women.
Osadchiy et al. (2019) [45]	Normal BMI: 28.95 (11.15) High BMI: 33.42 (10.83)	186	54.8	BMI ≥ 25 kg/m^2^, YFAS *	Resting state fMRI	The association between the centrality of VTA and YFAS was positive in females but negative in males. The association between the centrality of the ventrolateral prefrontal cortex and YFAS was positive in males but negative in females.
Gupta et al. (2018) [44]	Female: 29.84 (7.45) Male: 31.79 (10.62)	86	50	BMI ≥ 25 kg/m^2^	Resting state fMRI	Slow-4 signal in the right globus pallidus and bilateral putamen was associated with BMI in the female cohort, but not in the male cohort. Slow-5 connectivity between the left globus pallidus and substantia nigra with the bilateral posterior mid cingulate cortex and frontal cortical regions was negatively associated with BMI among females. Slow-5 connectivity between the left globus pallidus and substantia nigra and the medial frontal cortex was positively associated with BMI in the male cohort.

* Yale food addiction scale.

**Table 3 nutrients-12-01942-t003:** Brain taste response study sample characteristics and findings.

Paper	Age (years), Mean (SD)	Sample Size	Female%	Obesity Metrics	Neuro-Imaging Modality	Pertinent Findings
Cornier et al. (2015) [56]	Obese resistant: 30.4 (2.6) Obese prone: 30.2 (3.7)	49	49.0	Obesity proneness defined by history of diet and weight-gain	Task fMRI (cue anticipation task)	Obese-prone and -resistant males had greater neuronal response to the sucrose-associated visual cue in the right caudate nucleus relative to women.
Geliebter et al. (2013) [57]	Female: 35(6.9) Male: 35(9.0)	31	45.2	BMI ≥ 30 kg/m^2^	Task fMRI (cue reactivity task)	Male participants with obesity portrayed brain activation in response to high energy dense auditory food cues (rel. to low energy dense) in supplementary motor areas (precentral gyrus) in a sated state. Female participants showed activation in response to high energy dense auditory food cues (rel. to low energy dense) in parahippocampal gyrus in a fasted state.
Atalayer et al. (2014) [58]	Female: 35 (6.9) Male: 35 (9.0)	31	45.2	BMI ≥ 30 kg/m^2^	Task fMRI (cue reactivity task)	In a sated state, men demonstrated greater connectivity in the amygdala than women, while women displayed greater connectivity in the angular gyrus and precentral gyrus than men. In a fasted state, men displayed greater connectivity in the supplementary motor area, precentral gyrus, precuneus, cuneus, while women had greater connectivity in the inferior frontal gyrus.
Haase et al. (2011) [15]	Female: 21.94 (1.9) Male: 22.25 (2.7)	21	57.1	N/A	Task fMRI (cue reactivity task)	Men had greater brain activation decreases than women in response to all four tastes in the middle frontal gyrus, insula, and cerebellum when changing from a hunger to satiety state. Men had greater activational changes relative to women in reaction to sucrose, citric acid, and caffeine in the inferior frontal gyrus; sucrose and NaCl within the parahippocampal gyrus, entorhinal cortex, perirhinal cortex, and amygdala; and sucrose within the dorsal striatum (caudate, putamen) and posterior cingulate.
Wang et al. (2016) [59]	46.5 (9.3)	13	61.5	1991 NIH guidelines for obesity surgery *	Task fMRI (cue anticipation task)	Participants had decreased response in the reward center (including NAcc, caudate nucleus, VTA, OFC, and prefrontal cortex) in response to sucrose after gastric bypass and increased response in the same region in response to NaCl.

* NIH consensus development program, office of disease prevention. Gastrointestinal surgery for severe obesity. http://consensus.nih.gov/1991/1991gisurgeryobesity084html.htm. Mar 1991.

**Table 4 nutrients-12-01942-t004:** Brain neurotransmission study sample characteristics and findings.

Paper	Age (years), Mean (SD)	Sample Size	Female%	Obesity Metrics	Neuro-Imaging Modality	Pertinent Findings
Adams et al. (2004) [82]	Female: 47.4 (19.6) Male: 45.4 (20.1)	52	42.31	N/A	PET [^18^F]altanserin 5-HT2A	No sex differences in 5-HT2A binding Positive correlation between 5-HT2A binding and BMI
Erritzoe et al. (2010) [89]	35.7(18.2)	60	38.33	Overweight BMI > 25 kg/m^2^ Obese BMI ≥ 30 kg/m^2^	PET [^11^C]DASB 5-HTT	Negative correlation between 5-HTT binding and BMI Females > males in midbrain 5-HTT binding No interactions between BMI and gender
Koskela et al. (2008) [90]	25.42 (1.29)	32 (16 mono-zygotic twin pairs)	50	N/A	SPECT [^123^I]nor-β-CIT 5-HTT	Female, but not male, monozygotic twins with higher BMIs had higher 5-HTT binding in the hypothalamus and thalamus than their leaner co-twins
Wang et al. (2011) [106]	Lean: 37.5 (5.9) Obese: 38.9 (7.3)	20	40	Severely obese BMI > 40 kg/m^2^	PET [^11^C]raclopride D2/D3	Obese < controls in striatal D2/D3 receptor binding; positive correlation between D2/D3 receptor binding and BMI No sex differences
Burghardt et al. (2015) [121]	Lean: 51.43 (11.18) Obese: 52.43 (8.98)	14	0	ND Obese group mean BMI = 37.96 (1.83) kg/m^2^	PET [^11^C]carfentanil MOR	Obese < lean in MOR binding Partial recovery of MOR binding after restricted-calorie intervention in sample with obesity
Joutsa et al. (2018) [122]	Morbidly obese: 41.8 (10.3) Controls for morbidly obese: 44.9 (12.9) BED: 49.4 (5.1) Controls for BED: 43.1 (11.4)	56	100	ND Obese group mean BMI = 40.7 (3.8) kg/m^2^ BED group mean BMI = 30.9 (6.6) kg/m^2^	PET [^11^C]carfentanil MOR	Morbid obesity and BED < controls in MOR binding No differences in MOR binding between morbidly obese group and BED group
Karlsson et al. (2015) [108]	Lean: 44.86 (12.88) Obese: 39.08 (10.74)	27	100	ND Obese group mean BMI = 41.89 (3.88) kg/m^2^	PET [^11^C]carfentanil MOR [^11^C]raclopride D2/D3	Obese < control in MOR binding No differences in D2/D3 receptor binding
Tuominen et al. (2015) [123]	Lean: 42.00 (13.20) Morbidly obese: 41.24 (9.17)	45	100	ND Morbidly obese mean BMI = 41.30 (4.14) kg/m^2^ mean fat percentage = 50.34 (3.69)	PET [^11^C]carfentanil MOR [^11^C]raclopride D2/D3	Obese < control in MOR binding Positive correlation between MOR and D2/D3 receptor binding in ventral striatum in control, but not morbidly obese group

ND: Not defined.

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
