# Peer review of "Neuroimaging of Sex/Gender Differences in Obesity: A Review of Structure, Function, and Neurotransmission"

_nutrients, 2020, doi:10.3390/nu12071942_

Round 1
Reviewer 1 Report
I congratulate the authors on their comprehensive and well written review on sex/gender brain differences in obesity. The manuscript is a pleasure to read and very concise - the conclusions are supported by the presented literature.
On reading the manuscript, some questions – or rather comments and suggestions – came to my mind, which I would like to read the authors opinion on.
Limitations
- Although the authors mention this briefly, I think it would be good to explicitly highlight the limitation that very many studies in the context of eating behaviour are conducted in exclusively female samples. This severely hampers a balanced view on the generalisability of these studies’ findings.
- Many studies in the context of obesity do not include a proper assessment of the presence of eating disorders, leaving the possibility open that results ascribed to obesity might rather be explained by the presence of e.g. an undiagnosed BED.
- Recent studies re-investigated sex/gender differences in brain structure (e.g. Ritchie et al. Cer Cor 2018; Kijonka et al. Front Neurosc. 2020). The main findings were that many of the “known” differences between sexes/genders disappear or are less pronounced when total brain volume (TBV) or intracranial volume are taken into account. I would suggest to include this as a limitation, since many studies with a focus on obesity do not account for sex/gender differences in TBV or ICV.
Interpretation
- In the discussion, differences in brain volume are interpreted as atrophy. While this might be the case, volume differences might also represent neuroplasticity or a predisposition, especially in younger cohorts. I would suggest to carefully check causal wording and at least acknowledge other potential interpretations of structural differences.
- Aging is differentially associated with hormonal drifts in men and women. For women, menopause is associated with both, dramatic changes in sex hormone levels and balance and a change in obesity prevalence. This is normally not the case in men. I would suggest to include this aspect in the discussion.
- Neurotransmission: I would like to encourage the authors to discuss the implications of the fact that some of the tracers are displaceable by the agonist and others are not. What does this mean for the interpretation, e.g. of results obtained with raclopride?
- For dopamine, general sex/gender differences have been described (e.g. Pohjalainen et al. 1998; Kaasinen et al. 2001). How would this relate to obesity-associated sex/gender differences?
Potentially useful studies to include
Atalayer et al. NeuroImage 2014, functional connectivity & food cues
Minor:
Introduction:
Please add the unit to BMI, i.e. kg/m^2
Please elaborate: How exactly can neuoimaging be used to explain compulsive eating behaviours?
Please move the definition/explanation regarding the use of sex/gender to an earlier instance in the introduction.
Figure 1. Please add that the structural differences relate to GM early on in the caption.
Please check the validity of the term ‘neurological’ throughout the manuscript. I had the impression that sometimes the term ‘neural’ would have been more appropriate.
Reviewer 2 Report
The review proposed by Kroll et al. explores the neural and behavioral differences in obesity between sexes/genders. The review is of interest and very well written and referenced.
However, I have one general comment about the definition of obesity (BMI based) and associated complications. Total fat is not the appropriate factor to consider when studying metabolic complications in obesity but rather the ratio between subcutaneous fat (SAT) and visceral fat (VAT). Women before menopause have generally more SAT than VAT which protect them for metabolic complications, as opposed to men. These differences disappear after menopause which suggest a key role of sex hormones and especially estrogen. These differences should be clearly stated at the beginning of the review (introduction). In addition, the different patterns observed in women and men behavior in weight loss interventions could be more societal that physiological.
In addition, one specific comment that I believe should be added to the already extensive review is the sex differences in insulin/leptin brain activity to food addiction. Indeed, several studies showed that males and females are differently sensitive to insulin/leptin secretion. (Differential Sensitivity to Central Leptin and Insulin in Male and Female Rats. PMID: 12606509; Sex differences in metabolic regulation and diabetes susceptibility, https://doi.org/10.1007/s00125-019-05040-3). This should be discussed in the review.
